# Dual-Purpose of the Winged Bean (*Psophocarpus tetragonolobus* (L.) DC.), the Neglected Tropical Legume, Based on Pod and Tuber Yields

**DOI:** 10.3390/plants10081746

**Published:** 2021-08-23

**Authors:** Sasiprapa Sriwichai, Tidarat Monkham, Jirawat Sanitchon, Sanun Jogloy, Sompong Chankaew

**Affiliations:** Department of Agronomy, Faculty of Agriculture, Khon Kaen University, Khon Kaen 40002, Thailand; sasiprapa.sr@kkumail.com (S.S.); tidamo@kku.ac.th (T.M.); jirawat@kku.ac.th (J.S.); sanun@kku.ac.th (S.J.)

**Keywords:** winged bean, pod yield, tuber weight, nutrient content, protein content

## Abstract

Winged beans (*Psophocarpus tetragonolobus* (L.) DC.) are grown as a vegetable legume crop in Thailand. All parts of the plant, including pods, seeds, leaves, flowers, and tubers are edible and are rich in protein and nutrients. Although the major consumption of winged bean is based on pod and tuber yields, only the people of Myanmar and Indonesia utilize winged bean tubers as food materials. The usefulness of the winged bean as an alternative crop for staple food and feed can shed some light on the impact of winged bean. Therefore, the evaluation of the dual purpose of the winged bean based on pod tuber yields is the objective of this study. In this study, ten-winged bean accessions—six accessions obtained from introduced sources and four accessions obtained from local Thai varieties—were laid out in randomized complete block design (RCBD) with three replications at the Agronomy Field Crop Station, Faculty of Agriculture, Khon Kaen University, Khon Kaen, Thailand from September 2019 to April 2020 and from October 2020 to April 2021. Data, including total pod weight, number of pods, pod length, 10-pod weight, and tuber weight were recorded, and the proximate nutrient and mineral contents in the tubers were also determined. The results revealed that the principal effects of year (Y) and genotype (G) were significant for total pod weight and the number of pods. Moreover, the Y × G interactions were principal effects upon the total pod weights and tuber weights. The results indicated that superior genotype and appropriate environmental conditions are key elements in successful winged bean production for both pod and tuber yields. The winged bean accessions W099 and W018 were consistent in both experimental years for pod and tuber yields at 23.6 and 18.36 T/ha and 15.20 and 15.5 T/ha, respectively. Each accession also proved high in tuber protein content at 20.92% and 21.04%, respectively, as well as significant in fiber, energy, and minerals. The results suggest that the winged bean accessions W099 and W018 can be used for dual-purpose winged bean production in Thailand.

## 1. Introduction

The winged bean (*Psophocarpus tetragonolobus* (L.) DC.) is an underutilized tropical leguminous species, classified in the family of Fabaceae and subfamily of Papilionoideaeis [1]. Winged bean is an important tropical vegetable legume with high nutritional value [2], that can be grown in humid, tropical countries such as Indonesia, Malaysia, Bangladesh, and Thailand [3,4]. The winged bean can be cultivated in all of Thailand’s provinces, and produces edible pods, seeds, leaves, flowers, and tuberous roots that are rich in protein. As a tropical legume, its seeds contain high amounts of protein and oil [5], and it is often referred to as the ‘soybean of the tropics’ [6,7]. Young pods of the winged bean are consumed in raw, steamed, boiled, stir-fried, or pickled forms. In Southeast Asia, young pods are generally cooked in a variety of ways or consumed as a side dish or salad. In Myanmar, the crop is also popularly grown specifically for its young tuberous roots. The immature pods contain 1% to 3% protein, as well as several vitamins and minerals [8]. The winged bean’s mature seeds contain protein levels of 28% to 45% [9], oil of roughly 14% to 19%, and carbohydrates of 34% to 40% [10]. Moreover, its raw tubers contain 12% to 19% protein and 1% to 4% fat [10].

The immature wing bean pods represent its major form of consumption, as they are rich in minerals and vitamins, particularly vitamin A [11]. In Thailand, the winged bean is an underutilized crop that Thais consume in a variety of ways. Immature pods are used for salads, soups, and direct consumption. Tuber roots are typically roasted or boiled and consumed directly or made into confectionaries. In Thailand, winged beans are generally grown on smaller commercial scales that supply young pods to local markets. The summer market price of winged beans ranges between USD 8 to 10 per Kg. However, while the winged bean has the potential to become an important economic food crop in Thailand, very little research has been conducted on it over the past few decades.

All the winged bean cultivars grown in Thailand are either landraces or selections from landraces. Seeds that growers use for cultivation may also be from other provenances or geographic regions that have perhaps traveled with a farmer’s relatives [12]. Additionally, no genetic improvement program for the crop currently exists. Moreover, the yield potential was not evaluated, nor was an assessment of winged bean accessions on commercially desired traits such as pod length, pod tenderness, taste, and pod color, which directly affect consumer preferences.

In the Mandalay region of Myanmar and the Papua province of Indonesia, the tuber of the winged bean represents a staple food, again, due to its high protein contents. The crude protein yield of the winged tuber was estimated to be at least 300 to 600 kg/ha [13]. In Thailand, few areas, such as some remote villages in the Photharam district of Ratchaburi province consume the winged bean tuber, as research on it and its consequent popularity have not been established. Furthermore, winged bean research has focused primarily on pod yield only. After the summer’s pod production, the winged bean dries up and is removed for the next rotation of crops, such as corn, at which time tubers are discarded and not utilized. Eagleton [13] reported their highest tuber yield at 2629 kg dry matter/ha, suggesting the necessary utilization of the winged bean’s productivity and value. Note that not all accessions can produce tubers. Hildebrand [14] determined that only 38 out of 189 genotypes are capable of producing tubers.

Today’s ever-growing global population has increased the demand for animal-sourced feed, particularly in developing countries [15,16]. Today’s food-feed materials compete worldwide as human food and livestock feed, and commonly contain the same ingredients. Protein and carbohydrate sources, such as maize, cassava, and bean have fueled high feed prices, which in turn has generated new interest in alternative N sources for livestock feed. Therefore, food-feed production systems must be integrated for livestock-crop production. In this type of crop system, farmers would harvest produce for human consumption, whereas crop residue or byproducts would be utilized as feed for livestock [17]. Recent research on the potential of winged bean production has provided a framework for continued study of the winged bean as an alternative, staple crop for both food and feed, particularly within low input cropping systems of the tropics and subtropics. The objective of this study, therefore, was the evaluation of the dual purpose of the winged bean based on pod tuber yields.

## 2. Results

### 2.1. Genotype Response and Environment

The combined analyses of variance for total pod weight, number of pods, pod length, 10-pod weight, and tuber weight of ten winged bean accessions under two experimental years revealed that the principal effects (year (Y), genotype (G), and Y × G interaction) were significant (Table 1). This experimental year significantly affected the total pod weight and the number of pods with a 95% confidence level, whereas the genotype effects were statistically significant on the total pod weight, the number of pods, and pod length at a 99% confidence level. Additionally statistically significant was the Y × G interaction’s effect on total pod weight and tuber weight at a 95% confidence level (Table 1). Our results found significant effects upon all traits, except 10-pod weight, due to its uniformity within the marketplace. 

### 2.2. Potential of Pod-Related Traits and Tuber Yield Production of the Winged Bean

Variations were observed in pod-related traits in both experiment years, except for the 10-pod weights (Table 2). In the 2019 experiment, the pod yields of 10 winged bean accessions ranged from 23.6 T/ha (W099) to 5.58 T/ha (W077). Four winged bean accession including the W099, W018, W061, and W048 accessions, showed high yields of 23.64, 18.36, 15.82, and 15.06 (T/ha), respectively. The same accessions also produced a high number of pods. The W099 accession was highest in pod number at up to 39,354 × 102 pods/ha (Table 2). The 10-pod weights ranged from 62.97 to 59.85 g, and pod lengths ranged from 14.51 to 13.11 cm (Table 2). In the 2020 experiment, the W099, W001, W018, and W061 accessions generated the highest pod yields of 11.7, 11.18, 8.44, and 6.42 T/ha, respectively. These same accessions also presented the highest number of pods. Similar to the results of the 2019 experiment, the W099 accession achieved the highest number of pods at up to 15,880 × 102 pods/ha (Table 2). The 10-pod weight ranged between 77.89 to 65.15 g, and the pod lengths ranged from 14.54 to 13.06 cm (Table 2). Our assessment of the pod-related traits indicated that four winged bean accessions (W099, W001, W018, and W061) showed the highest yield performance in pod production.

Variations were also observed in tuber yields of both experiment years, in which the winged bean accessions W048, W148, W007, W018, and W099 produced yields greater than 15 T/ha in the 2019 experiment (Table 3). The 2020 experiment, however, produced lower tuber yields from the W018, W001, and W099 accessions at just over 9 T/ha (Table 3). Only the W018 and W099 accessions were consistent in tuber yield in both experiment years. The W018 and W099 winged bean accessions produced significantly higher pod and tuber yield potentials (Table 2 and Table 3), deeming them suitable for dual-purpose winged bean production (Figure 1).

### 2.3. Proximate Analysis of Nutrients and Minerals of Winged Bean Tuber

The proximate compositions of nutrient and mineral contents in the tuber of each winged bean accession were statistically significant at a 99% confidence level (Table 4 and Table 5): the values of crude fat ranged from 1.13% (W148) to 0.26% (W031); the values of crude fiber ranged from 4.07% (W018) to 2.37% (W099); the values of neutral detergent fiber ranged from 32.38% (W061) to 15.29% (W148); the values of acid detergent fiber ranged from 8.98% (W018) to 5.77% (W005); the values of gross energy ranged from 16,241 g/J (W005) to 15,810 g/J (W099); the values of ash ranged from 3.03% (W061) to 2.48% (W005); and the values of crude protein ranged from 25.59% (W061) to 20.41% (W005) (Table 4). The results of the proximate composition analysis of winged bean tubers in this study suggests its suitability for consumption or use as an alternative source for animal feed, due to its high protein (up to 25.59%) and energy (up to 16,264 J/g) contents (Table 5).

The following variabilities were observed in the mineral compositions of winged bean tubers: W001 produced the highest content of Fe (105.19 mg/kg) and B (42.63 mg/kg); W007 was highest in Fe as W001 (102.52 mg/kg) and Cu (14.79 mg/kg); W018 had the highest content of Mg (0.31%), Ni (2.70 mg/kg), and Co (0.35 mg/kg); W048 produced the highest level of P (0.34%) and K (0.62%); W055 was highest in Co (0.33 mg/kg); W061 had the highest N (3.51%), Ca (0.16%), S (0.12%), and Mn (32.03 mg/kg) contents; and W148 had the highest content of Zn (47.36 mg/kg) (Table 5). The results indicate that each winged bean accession is a significant source of mineral composition, which further suggests its use as a nutritional supplement.

## 3. Discussion

### 3.1. Genotype Response and Environment

This study investigated ten winged bean accessions of Thai and introduced sources sown into the Agronomy Field Crop Station, Faculty of Agriculture, Khon Kaen University, Khon Kaen, Thailand. Planting dates were determined by end of the rainfall, as high amounts of humidity and moisture cause plant stunning, as well as the occurrence of plant diseases [18]. In Thailand, bimodal rain creates two possible sowing dates: at the first drop of rain in June, and at the end of the rain season, mid-September to early October (Figure 2). A June planting may result in plant stress from pests, diseases, and water. Moreover, the winged bean plant is long-staying in the field typically until flowering, due to its photoperiod-sensitivity [13]. Within both periods of this study, all winged bean accessions were flowering on 17–22 November (data not shown). 

In the study, herein, pod yield of the winged bean was affected by year (Y), genotype (G), and Y × G interaction, where Y was a major proportion of variation (Table 1). Our results agreed with those of Stephenson [19], in which a genotype by environment interaction (GxE) affected the yield of the winged bean. While Y, G, and Y × G affected pod yields, the two most consistent high pods yields were achieved by the W099 and W018 accessions (Table 2); which demonstrates their ability to grow in different environmental conditions. As mentioned earlier, Thailand has no genetic improvement program for the winged bean. The cultivars grown in Thailand are either landraces or selections from landraces with high pod yields and eating qualities. Varieties demonstrating the best yield performances typically result in their migration from place to place [12]. 

The total pod weights and number of pods were greater in the 2019 experiment versus those of the 2020 experiment, perhaps due to the longer duration of the vegetative phase, 66 ± 3 days and 48 ± 2 days, respectively. Other external factors affecting pod yield, particularly in the 2020 experiment, were insect pests, such as flower bud thrips (*Megalurothrips sjostedti* (Trybom)) and the bean pod borer (*Maruca vitrata*), which were a major cause of flower and pod defoliation, that further led to low pod numbers and pod weights in 2020 (Table 2). Khan [18] and Reddy [20]; reported that a wide range of insects belonging to Lepidoptera cause extensive flower damage, causing them to dry out and fall prematurely without forming pods. Infested pods are scarred and deformed which adhere to the flowers [21]. Control of flower bud thrips and the bean pod borer [22,23,24,25,26,27] is necessary for winged bean production before the insect appears. However, excessive use of chemicals on fresh pods is of major concern. 

### 3.2. Potential of Pod-Related Traits and Tuber Yield Production of the Winged Bean

Fresh green pods are a major factor in the consumption of winged beans in Thailand and are evident by their green color, soft feel, and non-bitter taste [12]. Within the present study, all winged bean accessions were preliminarily selected for these traits before future yield trials were conducted (Table 6). Notably, both 10-pod weight and pod length (63.03 to 71.67 g and 13.79 to 13.89 cm, respectively) were not characteristic variables (Table 1 and Table 2), as they are considered proximate norms within the Thai winged bean marketplace. 

Winged bean production includes a sensitive photoperiod that limits year-round production causing a fluctuation in the prices of the winged bean. Growers normally sow winged bean seeds at the beginning of the rainy season (the first or second rain in a bimodal rain system). The plant typically flowers in October through November, creating a two to three-month harvest. Pod production then decreases in March. Upon the completion of the harvest, the plant dries, summer leaf defoliation takes place, and growers no longer attempt irrigation, as pods are no longer produced. Any remaining winged bean plants and tubers are utilized only as cover crops or tilled into the soil as manure. 

As mentioned earlier, only some villages in the Photharam District, Ratchaburi province of Thailand establish and promote the consumption of winged bean tubers. Because tuber yields are only achieved after digging up the plant, these farmers sow winged bean seeds from July to August and harvest the tubers once at six months after planting, from January to February [13]. Tubers are cut, washed, and then boiled at least for two hours. Boiled winged bean tubers can fetch a price of up to USD 60 per Kg. However, the winged bean tuber’s popularity remains fixed in other areas of Southeast Asia, like Papua New Guinea and Myanmar [13].

As the first known study of winged bean tuber yield in Thailand, our results demonstrated that some winged bean accession gave a tuber yield of more than 15 T/ha (Table 3), significantly higher than the 11.7 T/ha peak yield reported by Khan [28]. Our two winged bean accessions, W018 and W099, were consistent highest in tuber yield (Table 3), total pod weight, and the number of pods (Table 2). Additionally, only some accessions are capable of producing tubers [14,29,30]. Moreover, few accessions have been regarded as high in pod and tuber yields, or ‘dual-purpose’ [13]. The W018 and W099 winged bean accessions produced successful pod and tuber yields, are consequently classified as dual-purpose winged accessions. However, before recommending these accessions to winged bean growers throughout the country, a multi-location yield trial is required.

### 3.3. Proximate Analysis of Nutrients and Minerals of Winged Bean Tuber

Interestingly, the nutrient content analysis in the present study shows high protein contents of up to 25.59% (W061) (Table 4). The previous study by Adegboyega [15] reported that the winged bean accession Tpt42 contained 19.07% tuber protein. The W099 and W018 accessions herein produced tuber protein contents of 20.92% and 21.04%, respectively (Table 4), which are higher than those of typical tuber crops, like cassava [31,32]. Our study also considered the fiber, energy, and minerals contents of the winged bean tubers (Table 4 and Table 5). 

Because of the low consumption and utilization of winged bean tubers in Thailand, alternative uses, such as feed material for livestock may also be considered. Recent results on the potential of winged bean production have provided a framework for continued study into the usefulness of the winged bean as an alternative crop for staple food and feed, particularly within low input cropping systems in the tropics and subtropics as food-feed production systems [17]. The variation and high amounts of nutrient and mineral contents in winged bean tubers suggest its potential as food security and nutritional modification in tropical agriculture [15].

## 4. Materials and Methods

### 4.1. Plant Material

Ten accessions of winged bean (*P. tetragonolobus* (L.) DC) were used in this study. Six accessions were obtained from introduced sources from the Gene bank of NIAS—Japan provided by Dr. Prakit Somta, Kasetsart University, Thailand and four accessions were obtained locally in Thailand (Table 6). All accessions were grown in 2019 and 2020 for genetic diversity study in the Agronomy Field Crop Station, Khon Kaen University, Khon Kaen, Thailand. A visual selection of young pod characteristics, such as pod number, softness, color, and non-bitter taste was conducted. 

### 4.2. Field Experiment

The 10 winged bean accessions were laid out in a randomized complete block design (RCBD) with three replications at Khon Kaen University’s Agronomy Field Crop Station, Faculty of Agriculture from September 2019 to March 2020 and from October 2020 to April 2021. The experimental plots were 5 × 1 m, spaced 1 m between rows and 0.5 m between plants, totaling ten plants per plot, with a distance of 2 m between plots within each row. A 2 m high net constructed of bamboo and nylon was used for support. Fertilizer was initially applied at the rate of 14.06 kg/ha (N_2_-P_2_O_5_-K_2_O) at 21 days after planting (DAP). A second fertilizer application was applied at the rate of 18.75 kgN_2_/ha, 37.50 kgP_2_O_5_/ha, and 18.75 kgK_2_O/ha at two months of age. Manual weed control was practiced regularly during the growing period. Plants were watered regularly, and disease and pest control were conducted as required throughout the growing period. 

### 4.3. Data Collection

Data including total pod weight (g), number of pods, 10-pod weight (g), and pod length (cm) were recorded. Pod-related traits were recorded at three-day intervals during the two months of pod production. Tuber weights were recorded at eight months. Tubers were removed from the plant, and their fresh weights were immediately recorded. The fresh tubers of the winged bean accessions in each plot (approximately ten tubers) were immediately sub-sampled, washed in tap water, and then sliced into small chips and oven-dried at 50–55 °C for 48 h, or until a constant weight was achieved via a tray drier (EQ-04SW, Leehwa Industry Company, Kyongbuk, Korea). The chipped samples were ground into a powder with a grinder (Standard EM-11, Sharp Thai Company Limited, Bangkok, Thailand). The powdered samples were then sieved through a 1.0 mm mesh screen that helped to ensure that the sample for chemical analysis was representative. The samples were later used for laboratory analyses. 

Nutritional analysis was performed at the Animal Nutrition Unit, Department of Animal Science, Khon Kaen University, Khon Kaen, Thailand. 

Crude fat or ether extract (EE), crude fiber (CF), and ash were determined by the Weende method according to the AOAC [33] with slightly modified were briefly described below. 

Determination of EE: the 5.0 g powdered sample was extracted in 100 mL diethyl ether and shaken it for 24 h in an orbital shaker. The filtrate was collected in the same flask after it was equilibrated with 100 mL diethyl ether and again shaken for 24 h. The ether was dried in an oven at 60 °C for 30 min after being concentrated to dryness in a steam bath. The weight of ether extract was determined by difference and calculated as a percentage of the weight of sample analyzed [34] thus:(1)$$\mathrm{Crude}\;\mathrm{fat}\;(\%)=\frac{\mathrm{Weight}\;\mathrm{of}\;\mathrm{flask}\;\mathrm{with}\;\mathrm{fat}\;-\;\mathrm{weight}\;\mathrm{of}\;\mathrm{empty}\;\mathrm{flask}\;}{\mathrm{Weight}\;\mathrm{of}\;\mathrm{sample}\;}\times 100\;$$

Determination of CF: the 5.0 g powdered sample was processed with 100 mL of 1.25% H_2_SO_4_ for half an hour and filtered with pressure. The remaining residue was then washed with hot water. This process was repeated on the residue by using 100 mL of 1.25% NaOH sol. The remaining filtrate was dried at 100 °C. It was subsequently incinerated in a muffle furnace at 550 °C for 5 h. The weight of the fiber was determined by difference and calculated as a percentage of the weight of sample analyzed [35] thus:(2)$$\mathrm{Crude}\;\mathrm{fiber}\;(\%)=\frac{\;\left(\mathrm{Weight}\;\mathrm{of}\;\mathrm{crucible}\;+\;\mathrm{Ash}\right)-\;\mathrm{Weight}\;\mathrm{of}\;\mathrm{crucible}+\mathrm{sample}\;\mathrm{after}\;\mathrm{washing},\;\mathrm{boiling}\;\mathrm{and}\;\mathrm{drying}}{\mathrm{Weight}\;\mathrm{of}\;\mathrm{sample}\;}\times 100\;$$

Determination of ash: A porcelain crucible was dried at 105 °C for 1 h, and then the 5.0 g of powdered sample was placed in the crucible. The crucible with plant samples was ashed first at 250 °C for an hour, followed by ashing at 550 °C for five hours in a muffle heating system. The sample was then cooled in a desiccator. The weight of the ash obtained was determined by difference and calculated as a percentage of the weight of sample analyzed [35] thus:(3)$$\mathrm{Ash}\;(\%)=\frac{\left(\mathrm{Weight}\;\mathrm{of}\;\mathrm{crucible}\;+\;\mathrm{Ash}\;\right)-\;\mathrm{Weight}\;\left(g\right)\;\mathrm{of}\;\mathrm{empty}\;\mathrm{crucible}\;}{\mathrm{Weight}\;\mathrm{of}\;\mathrm{sample}\;}\times 100\;$$

Crude protein (CP) was determined via the Kjeldahl method according to the AOAC [33] as well. Briefly, about 10 g of potassium sulfate and 0.5 g of copper sulfate were added to flask 25 mL of concentrated sulfuric acid was added. The flask was plays on digestion chamber and heated gently to boil until contents were clear and allowed the liquid to cool and diluted with 200 mL of distilled until all ammonia was passed over and was received over standard sulfuric acid, which was then back titrated with standard NaOH in order to determine the amount of standard acid used to neutralize the ammonia evolved from digested material. Similarly, a blank sample was run.
(4)$$\mathrm{Total}\;\mathrm{nitrogen}\;(\%)=\frac{1.4\;\left(B-A\right)N}{\mathrm{Weight}\;\mathrm{of}\;\mathrm{sample}\;}\;\;$$
Crude protein = total nitrogen *×* 6.25(5)
where, B = volume of N/10 NaOH for blank, A = volume of N/10 NaOH used for sample, and N = normality of standard NaOH.

Neutral detergent fiber (NDF) and acid detergent fiber (ADF) contents were determined by detergent analysis according to Van Soest [36].

NDF was calculated using the following formula,
(6)$$\mathrm{NDF}\;(\%)\;=\frac{\left(\mathrm{Weight}\;\mathrm{of}\;\mathrm{crucible}\;+\;\mathrm{Fiber}\;\mathrm{content}\right)\;-\;\mathrm{Weight}\;\mathrm{of}\;\mathrm{empty}\;\mathrm{cricible}\;}{\mathrm{Weight}\;\mathrm{of}\;\mathrm{sample}\;}\;\times \;100\;$$

ADF was calculated using the following formula,
(7)$$\mathrm{ADF}\;(\%)=\frac{\mathrm{Weight}\;\mathrm{of}\;\mathrm{crucible}\;+\;\mathrm{Fiber}\;\mathrm{content}\;}{\mathrm{Weight}\;\mathrm{of}\;\mathrm{sample}\;}\times 100\;$$

An automatic adiabatic bomb calorimeter (AC500, Leco, St. Joseph, MI, USA) was used for gross energy (GE) estimation. Additionally, the nutrient contents of the winged bean tubers were determined. Atomic absorption spectrophotometry was employed to determine concentrations of total P, K, Ca, Mg, Fe, Cu, Zn, Ni, B, and Co through wet digestion (nitric-perchloric acid digestion) [37]. In brief, 5 mL of 65% HNO_3_ was added to the sample, and then the mixture was boiled gently for 30–45 min. After cooling, 2.5 mL of 70% HClO_4_ was added, and the mixture was gently boiled until dense white fumes appeared. Later, the mixture was allowed to cool, and 10 mL of deionized water was added followed by further boiling until the fumes were totally released [37]. Total N was determined via Kjeldahl method [33] as formula (4) and total sulfur through turbidimetry. Briefly, 3 mL of concentrated nitric acid was added to the samples or standards (0.5 g), the samples were left to stand for 15 min before closing the Teflon vessels and proceeding to digestion in the microwave oven. At the end of digestion, the vessel was cooled until a pressure of about 69 kPa was reached, then the lid was carefully removed. Next, the volume was adjusted to 50 mL with deionized water and S was quantified by ICP-AES [38].

### 4.4. Data Analysis

Data were analyzed using Statistix 10 of variance (ANOVA), and combined analysis was also performed. A least significant difference (LSD) at *p* < 0.05 was carried out for mean comparisons on the parameters measured among all accessions. 

## 5. Conclusions

Ten winged bean accessions were preliminarily evaluated across two production years for potential in pod and tuber yield as well as protein and nutritional content, referred to here as dual-purpose. The two accessions out of those, W099 and W018 were identified as high potential accessions for the two traits which are able to be source of food and feed stuff. However, further evaluation under diverse environments will depict availability of accession through the country.

## Figures and Tables

**Figure 1 plants-10-01746-f001:**
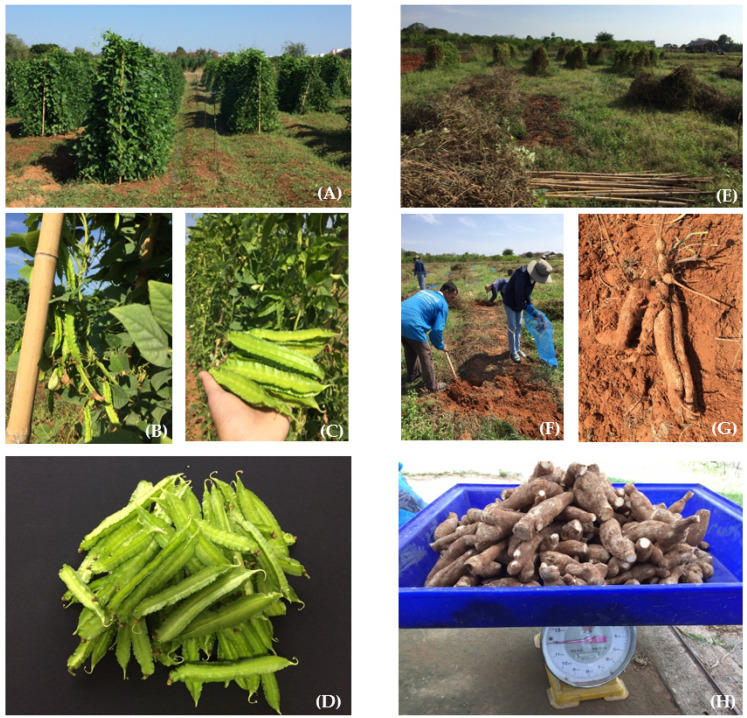
Dual-purpose of the winged bean (*Psophocarpus tetragonolobus* (L.) DC.) based on pod yield (**A**–**D**) and tuber yield (**E**–**H**).

**Figure 2 plants-10-01746-f002:**
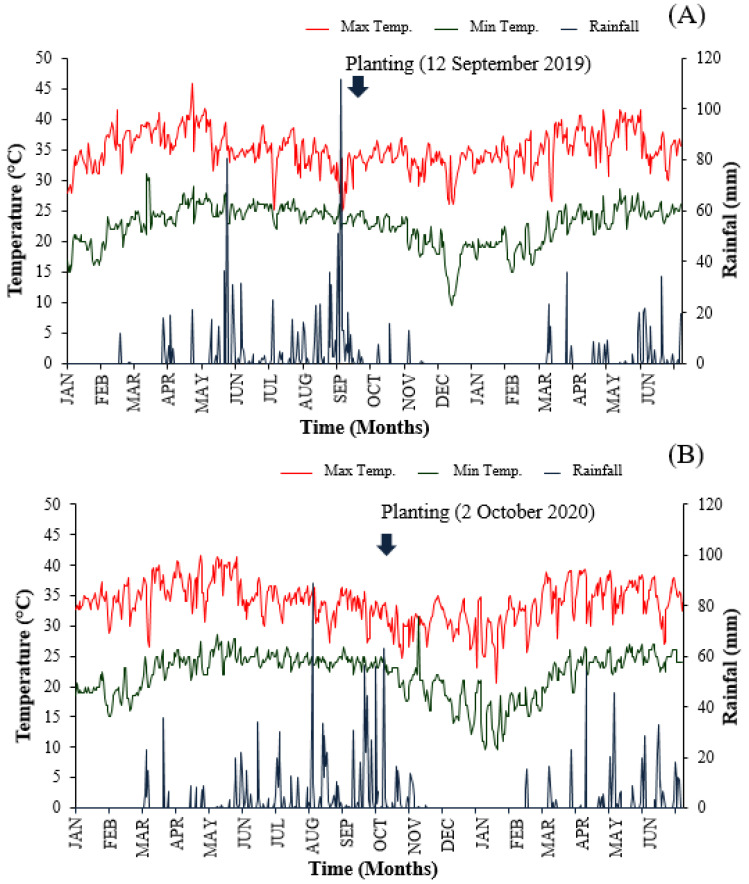
Average rainfall and the maximum and minimum temperatures during the 2019-2020 (**A**) and 2020-2021 (**B**) experiments in Northeast Thailand (Khon Kaen).

**Table 1 plants-10-01746-t001:** Mean squares of total pod weights, number of pods, 10-pod weights, pod lengths, and tuber weights of ten accessions evaluated across two years under rainy season conditions at Khon Kaen University in 2019 and 2020.

Source of Variation	df	Total Pod Weight	Number of Pods	10-Pod Weight	Pod Length	Tuber Weight
		(T/ha)	(Pod/ha)	(g)	(cm)	(T/ha)
Year (Y)	1	207.06 *	8,162,282 *	4.12	0.03	7.7
Rep. within Y	4	18.55	863,650	1263.34	0.31	6.38
Genotypes (G)	9	21.47 **	548,701 **	1101.94	1.33 **	12.68
Y × G	9	8.95 *	234,256	1239.4	0.35	15.56 *
pool error	36	3.49	137,609	1117.94	0.23	7.04
Grand Mean		4.83	760.67	71.93	13.81	7.33
CV(YearxRep)		89.26	122.17	49.41	4.01	34.45
CV(YearxRepxVariety)		38.71	48.77	46.48	3.49	36.19

* Significant at 0.05 probability level. ** Significant at 0.01 probability level.

**Table 2 plants-10-01746-t002:** Variability of winged bean total pod weight, number of pods, 10-pod weight, and pod length of ten accessions evaluated across two years under rainy season conditions at Khon Kaen University in 2019 to 2020.

Accessions	Total Pod Weight (T/ha)		Number of Pods (Pod/ha)		10-Pod Weight (g)		Pod Length (cm)	
	2019	2020	2019	2020	2019	2020	2019	2020
W001	10.94 b–d	11.18 ab	17,694 × 10^2^ b–d	14,280 × 10^2^ ab	61.84	77.16	14.11 a–c	14.49 ab
W005	7.22 cd	3.96 c–e	12,254 × 10^2^ cd	5493 × 10^2^ c–e	62.97	68.53	13.28 de	13.18 cd
W005	7.22 cd	3.96 c–e	12,254 × 10^2^ cd	5493 × 10^2^ c–e	62.97	68.53	13.28 de	13.18 cd
W007	5.58 d	6.10 b–e	8954 × 10^2^ d	7447 × 10^2^ c–e	61.76	62.79	13.93 a–d	14.54 a
W018	18.36 ab	8.44 a–c	32,186 × 10^2^ ab	11,367 × 10^2^ a–c	61.84	78.52	13.32 de	13.85 a–d
W031	12.72 d–d	1.22 e	20,500 × 10^2^ d–d	1867 × 10^2^ e	64.69	66.85	14.15 a–c	13.19 cd
W048	15.06 b	2.46 de	24,114 × 10^2^ a–d	3140 × 10^2^ de	65.37	69.92	14.51 a	14.00 a–c
W055	13.12 bc	5.02 c–e	22,686 × 10^2^ a–d	6107 × 10^2^ c–e	59.85	65.15	13.63 b–e	13.90 a–d
W061	15.82 b	6.42 b–d	28,406 × 10^2^ a–c	8833 × 10^2^ b–d	62.48	73.22	13.11 e	13.06 d
W099	23.64 a	11.7 a	39,354 × 10^2^ a	15,880 × 10^2^ a	67.43	76.68	14.40 ab	14.46 ab
W148	11.20 b–d	2.86 de	19,754 × 10^2^ b–d	3953 × 10^2^ de	62.13	77.89	13.46 c–e	13.65 b–d
Mean	13.37	5.94	22,590 ×10^2^	7837 × 10^2^	63.03	71.67	13.79	13.89
F-test	**	**	*	**	ns	ns	**	**
CV (%)	32.73	49.9	43.2	49.17	5.07	15.88	3.27	3.69

The different letter after mean within column showed significant different. ns Nonsignificant, * Significant at 0.05 probability level, and ** Significant at 0.01 probability level.

**Table 3 plants-10-01746-t003:** Tuber weights of ten winged bean accessions evaluated across two years under rainy season conditions at Khon Kaen University in 2019 and 2020.

Accessions	Tuber Weight (T/ha)	
	2019 Experiment	2020 Experiment
W001	12.00 b–d	10.20 a
W005	13.54 b–d	4.50 e
W007	16.46 a–c	6.80 cd
W018	15.54 a–c	10.23 a
W031	12.2 b–d	8.10 bc
W048	23.00 a	5.87 de
W055	7.96 cd	5.90 de
W061	6.14 d	8.13 bc
W099	15.20 a–d	9.07 ab
W148	17.44 ab	8.07 bc
Mean	14.00	7.69
F-test	*	**
CV (%)	37.91	12.42

The different letter after mean within column showed significant different. * Significant at 0.05 probability level. ** Significant at 0.01 probability level.

**Table 4 plants-10-01746-t004:** Proximate composition of winged bean tubers of ten accessions at Khon Kaen University in the 2019 experiment.

Accessions	EE (%)	CF (%)	NDF (%)	ADF (%)	GE (J/g)	Ash (%)	CP (%)
W001	0.29 ef	3.20 c	21.68 g	6.24 e	16,174 ab	2.96 b	21.66 c
W005	0.33 d	3.11 cd	24.46 d	5.77 g	16,241 a	2.48 e	20.41 e
W007	0.32 de	3.57 b	21.66 g	6.54 d	16,157 a–c	2.85 c	21.68 c
W018	0.31 de	4.07 a	22.91 f	8.98 a	15,990 cd	3.01 ab	21.04 d
W031	0.26 f	2.99 d	26.90 b	7.11 c	16,132 a–c	2.98 ab	22.11 b
W048	0.85 b	2.60 f	26.21 c	6.48 d	15,869 de	2.68 d	21.48 c
W055	0.88 b	3.02 d	16.62 h	7.00 c	15,845 de	2.85 c	20.71 de
W061	0.63 c	2.96 de	32.38 a	7.33 b	16,264 a	3.03 a	25.59 a
W099	1.13 a	2.37 g	24.20 e	5.92 f	15,810 e	2.62 d	20.92 d
W148	1.16 a	2.82 e	15.29 i	6.03 f	16,050 bc	2.62 d	20.73 de
F-test	**	**	**	**	**	**	**
CV (%)	2.75	2.26	0.44	0.81	0.49	0.59	0.72

The different letter after mean within column showed significant different. ** Significant at 0.01 probability level. EE = ether extract (crude fat), CF = crude fiber, NDF = neutral detergent fiber, ADF = acid detergent fiber, GE = gross energy, and CP = crude protein.

**Table 5 plants-10-01746-t005:** Mineral compositions of winged bean tubers of ten accessions at Khon Kaen University in 2019.

Accessions	N (%)	P (%)	K (%)	Ca (%)	Mg (%)	S (%)	Fe (mg/kg)	Mn (mg/kg)	Zn (mg/kg)	Cu (mg/kg)	Ni (mg/kg)	Co (mg/kg)	B (mg/kg)
W001	2.93 b	0.32 b	0.56 c	0.13 b	0.29 b	0.11 b	105.19 a	23.27 c	20.58 e	10.62 c	2.06 c	0.26 cd	42.63 a
W005	2.875 bc	0.26 e	0.53 d	0.10 e	0.21 g	0.10 c	77.85 d	19.30 d	22.78 cd	5.06 e	2.03 c	0.22 ef	29.74 d
W007	2.915 b	0.30 c	0.53 de	0.11 d	0.25 de	0.09 d	102.52 a	19.81 d	28.25 b	14.79 a	2.13 c	0.22 ef	21.84 g
W018	2.795 d	0.29 d	0.51 e	0.12 b	0.31 a	0.11 b	98.27 b	24.08 c	15.32 g	5.50 e	2.70 a	0.35 a	27.86 e
W031	2.85 cd	0.32 b	0.59 b	0.11 d	0.27 c	0.11 b	72.22 e	26.52 b	18.28 f	11.39 b	2.36 b	0.30 b	37.88 b
W048	2.625 e	0.34 a	0.62 a	0.09 f	0.20 h	0.09 d	64.54 f	17.69 e	18.46 f	5.14 e	1.88 d	0.21 f	36.46 c
W055	2.885 bc	0.32 b	0.58 b	0.12 c	0.24 e	0.10 c	71.05 e	20.25 d	21.15 de	10.60 c	2.02 c	0.33 a	23.37 f
W061	3.51 a	0.32 b	0.51 e	0.16 a	0.27 c	0.12 a	88.15 c	32.03 a	24.24 c	9.73 d	2.31 b	0.30 b	27.74 e
W099	2.84 cd	0.25 f	0.51 e	0.09 f	0.22 f	0.10 cd	76.71 d	24.18 c	19.54 ef	10.44 c	2.02 c	0.24 de	24.67 f
W148	2.475 f	0.30 c	0.53 d	0.085 f	0.24 d	0.09 d	87.48 c	23.57 c	47.36 a	5.06 e	2.34 b	0.28 bc	28.12 e
F-test	**	**	**	**	**	**	**	**	**	**	**	**	**
CV (%)	0.89	1.33	1.37	2.64	1.6	2.2	1.75	2.01	3.12	3.06	2.66	4.87	1.96

The different letter after mean within column showed significant different. ** Significant at 0.01 probability level.

**Table 6 plants-10-01746-t006:** The origins and sources of the ten selected winged beans used in this study.

Accessions No	Accessions Code	Fresh Pod Characters	Original and Sources
1	W001	Green color, non-bitter, short, and soft	Japan: Gene bank of NIAS—Japan
2	W005	Green color, non-bitter, short, and soft	Indonesia: Gene bank of NIAS—Japan
3	W007	Green color, non-bitter, short, and soft	Indonesia: Gene bank of NIAS—Japan
4	W018	Green color, non-bitter, short, and soft	Nigeria: Gene bank of NIAS—Japan
5	W031	Green color, non-bitter, short, and soft	Nigeria: Gene bank of NIAS—Japan
6	W048	Green color, non-bitter, short, and soft	Malaysia: Gene bank of NIAS—Japan
7	W055	Green color, non-bitter, short, and soft	Khon Kaen, Thailand
8	W061	Green color, non-bitter, short, and soft	Nakhon Si Thammarat, Thailand
9	W099	Green color, non-bitter, short, and soft	Nan, Thailand
10	W148	Green color, non-bitter, short, and soft	Trang, Thailand

## Data Availability

The data presented in this study are available on request from the corresponding author.

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
