# Peer review of "Dual-Purpose of the Winged Bean (Psophocarpus tetragonolobus (L.) DC.), the Neglected Tropical Legume, Based on Pod and Tuber Yields"

_plants, 2021, doi:10.3390/plants10081746_

Round 1

Reviewer 1 Report

This is an interesting manuscript on an important plant genetic resource. The document is well written and achieves interesting results. However, I think you need to make some corrections to be accepted:

The text is not correctly formatted. Tables that are horizontal should be placed vertically. Also, Table 1 should appear first in the text. Check the position of the rest of the tables.

Abstract: It does not report on the state of the art of the taxon under study and the objective is not clear.

Introduction: It offers a very good vision of the species under study. It informs about what is known about it, its importance and what is not known. Here the objective of the work appears clearly, something that is not observed in the abstract.

Material and Method: It is very complete and is very clarifying. Well raised.

Results: Researchers obtain a significant volume of information that is also of great interest.

Discussion: I believe that the discussion should have been organized into subsections. The same ones that the authors have used to present the Results could be used. If the authors make this change, the reading and comprehension will be much easier for the readers.

Conclusion: I consider that more than a conclusion it seems like a discussion. The authors should rethink it.

Author Response

Dear Reviewer, 

I revise manuscript as your recommend and clearly check through manuscript.

The response to reviewer  was attached file. 

Best regards 

Sompong Chankaew 

Reviewer 2 Report

The manuscript entitled “Dual-purpose of the winged bean [Psophocarpus tetragonolobus (L.) DC.]; the neglected tropical legume, based on pod and tuber yields” by Sasiprapa Sriwichai, Tidarat Monkham, Jirawat Sanitchon, Sanun Jogloy, and Sompong Chankaew describes the yield and the nutrient content of different cultivars of winged beans obtained in two subsequent years. The manuscript is generally well-written and clearly organised however I have some doubts with respect to its significance and originality.

Winged bean is unusual plants since the entire plant can be consumed. Moreover, it can be used not only as a food but also as a feed which make this plant valuable crop. Although it is grown and consumed in some restricted areas of Asia this plant is for sure worth studying. With respect to that the introduction is well-written and the aim and the reason for studying of winged beans is well explained. However, the results presented in this manuscript do not significantly improve the knowledge about winged beans. The analysed accessions vary in terms of yield, weight of pods and tubers, nutrient content which if fact is rather expected. Moreover, the differences for each accession in yield and other parameters obtained in each year are large even for W099 and W018 that have been chosen by Authors as the best ones. Again, this is obvious that in different growing seasons the obtained yield could be diversified due to several environmental factors. As mentioned by Authors in Thailand there is no programme for genetic improvement of winged bean so the presented in this manuscript results could be preliminary results for further genetic analysis of the selected accessions. Thus, although the manuscript is well-written its significance at this stage is rather low however it could be the beginning for deeper molecular analysis of this important and interesting crop.

I would suggest rewriting conclusion since based on the presented results it is not clearly visible that the selected by Authors accession are much better in terms of yield, nutrient content ect. than others. Therefore, for me these are rather preliminary results for further studies.  

 Small numbers of typos e.g., page 12 lines 265-266 – subscript, page 13 line 292 – should be total sulfur instead of “Total sulfur”.

Page 13, lanes 282-293 – please add some details about methods for determination of EE, CF, CP, NDF, ADF, nutrient contents, total N and total S.

Author Response

Dear Reviewer, 

I revise manuscript as your recommend and clearly check through manuscript.

The response to reviewer  was attached. 

Best regards 

Sompong Chankaew 

Round 2

Reviewer 1 Report

The authors have taken into account all my proposals and I consider that the text, which was already very good quality, has improved remarkably.

After the changes made, I consider that the manuscript is of sufficient quality to be published in Plants.